# Cellular-Based Therapies in Systemic Sclerosis: From Hematopoietic Stem Cell Transplant to Innovative Approaches

**DOI:** 10.3390/cells11213346

**Published:** 2022-10-24

**Authors:** Elisabetta Xue, Antonina Minniti, Tobias Alexander, Nicoletta Del Papa, Raffaella Greco

**Affiliations:** 1Hematopoietic and Bone Marrow Transplant Unit, San Raffaele Hospital, 20132 Milan, Italy; 2Department of Rheumatology, ASST G. Pini-CTO, 20122 Milan, Italy; 3Charité—Universitätsmedizin Berlin, Corporate Member of Freie Universität Berlin, Humboldt-Universität zu Berlin, and Berlin Institute of Health, Department of Rheumatology and Clinical Immunology, 10117 Berlin, Germany; 4Deutsches Rheuma-Forschungszentrum (DRFZ), an Institute of the Leibniz Association, 10117 Berlin, Germany

**Keywords:** systemic sclerosis, autoimmune diseases, cell therapies, hematopoietic stem cell transplantation, mesenchymal stem cells, adipose tissue grafting, skin fibrosis, digital ulcers

## Abstract

Systemic sclerosis (SSc) is a systemic disease characterized by autoimmune responses, vasculopathy and tissue fibrosis. The pathogenic mechanisms involve a wide range of cells and soluble factors. The complexity of interactions leads to heterogeneous clinical features in terms of the extent, severity, and rate of progression of skin fibrosis and internal organ involvement. Available disease-modifying drugs have only modest effects on halting disease progression and may be associated with significant side effects. Therefore, cellular therapies have been developed aiming at the restoration of immunologic self-tolerance in order to provide durable remissions or to foster tissue regeneration. Currently, SSc is recommended as the ‘standard indication’ for autologous hematopoietic stem cell transplantation by the European Society for Blood and Marrow Transplantation. This review provides an overview on cellular therapies in SSc, from pre-clinical models to clinical applications, opening towards more advanced cellular therapies, such as mesenchymal stem cells, regulatory T cells and potentially CAR-T-cell therapies.

## 1. Introduction

Systemic sclerosis (SSc) is a rare autoimmune systemic disease characterized by three main dysfunctions: autoimmune responses, vasculopathy and tissue fibrosis. The result is a progressive loss of the microvascular bed and the development of fibrosis of the skin and internal organs, including lung and gastrointestinal tract [1]. The first and most frequent symptom of presentation is Raynaud’s phenomenon, a direct sign of vascular dysfunction often complicated by digital ulcers (DU), one of the major burdens for SSc patients [2]. Another manifestation of vasculopathy is pulmonary arterial hypertension which, along with extensive skin and internal organ fibrosis, is responsible for increased disability and mortality in SSc patients [3,4].

The etiopathogenesis of the disease is complex, but substantial progress has been made in understanding the mechanisms involved. It is assumed that, in a genetically predisposed population, environmental factors lead to cytokines release, immune cell activation and connective tissue repair dysregulation [5,6,7]. Indeed, the fibrotic changes observed in SSc patients are likely due to a dysfunctional repair of connective tissues in response to injuries like oxidative stress. A wide range of cellular components are involved, mainly endothelial cells, mesenchymal cells, and T and B lymphocytes. Fibrosis is caused by excessive collagen and other extracellular matrix proteins’ production by myofibroblasts. The transition of endothelial cells to myofibroblast is induced by cytokines released by activated B and T cells (IL-6, IFN-alfa, IL-10, TGF-beta, PDGF) [8].

Currently, SSc is considered as incurable. The recent introduction of biologic disease-modifying drugs (DMARDs) and anti-fibrotic therapies has provided more specificity and effectiveness. However, such therapies usually only slow the disease progression and rarely reverse disease manifestations. Moreover, maintenance therapies expose patients to side effects, such as infections, and may be associated with a cumulative risk of co-morbidities [9]. The need to reset autoimmune pathogenic mechanisms before the appearance of extensive and irreversible damage has led to many efforts to develop cell-based therapies able to restore self-tolerance, from animal models to clinical practice [10]. Along with their tolerogenic properties, cell-based therapies showed an interesting potential as regenerative therapies able to repair tissues already damaged. 

Here, we review the available literature on cell-based therapies in SSc, from pre-clinical models to clinical applications, and approaches adopted in other autoimmune diseases potentially of interest for future investigations in SSc patients. 

## 2. Hematopoietic Stem Cell Transplantation

Hematopoietic stem cell transplantation (HSCT) has a consolidate role in treating hematologic malignancies. For more than 20 years, there has been a growing interest in its application for the treatment of autoimmune diseases, and as a consequence of three positive RCT, autologous HSCT is now considered a standard therapy in refractory SSc [11,12]. HSCT is a multistep procedure designed to replace the hematopoietic system of a patient with a new one derived from HSCs, which can be collected from a healthy donor (allogeneic HSCT) or from the same patient (autologous). The graft can undergo CD34 selection in order to enrich HSCs and potentially remove self-reactive lymphocytes or can be infused unmanipulated. The ablation of autoreactive immune cells is achieved through a conditioning regimen, which can be highly variable in intensity, ranging from fully to non-myeloablative, and generally includes anti-thymocyte globulin (ATG). A myeloablative regimen can be busulfan- or total body irradiation-based, whereas non-myeloablative regimen are mostly cyclophosphamide- or fludarabine-based. Intermediate intensity regimens are also adopted, especially in neurological autoimmune diseases [13,14,15,16,17].

The rationale of autologous HSCT for autoimmune diseases is to reset a dysfunctional, autoreactive immune system with the intent to restore a more naïve, self-tolerant one [18,19]. Several mechanisms that might participate in immunologic reset have been described, including thymic reactivation [20,21] with a vast diversification of the TCR repertoire [21], expansion of regulatory T cells, [22,23] and restoration of a naïve B-cell compartment with re-emergence of regulatory B cells [21].

Among initial reports in SSc, [24,25] Farge and colleagues documented the safety and clinical benefit of autologous HSCT in 12 patients within a phase I-II study. Hematopoietic recovery was observed in all patients, and 8 out of 11 evaluable patients achieved significant clinical response [26]. After these promising initial results, three randomized–controlled studies have been conducted for SSc, comparing autologous HSCT to standard immunosuppressive therapies. The phase II ASSIST trial randomized nineteen SSc patients to receive either non-myeloablative, unmanipulated HSCT or monthly cyclophosphamide. All patients in the HSCT arm showed both cutaneous and pulmonary improvements, compared to lack of significant long-term benefit after standard immunosuppression with cyclophosphamide. Notably, the authors suggested that interstitial lung disease, long considered as being irreversible, might be at least partially reversed after HSCT, with sustained improvements observed years after transplant [15]. In the phase III ASTIS trial, 156 patients were randomized to receive either non-myeloablative, CD34-selected HSCT or standard immunosuppression with monthly cyclophosphamide pulse. Despite higher early treatment-related mortality in the HSCT arm (10% versus none, mainly attributed to SSc-related cardiac dysfunction), event-free survival and overall survival was superior to cyclophosphamide during follow-up [16]. Finally, in the SCOT trial, 75 patients received either myeloablative, total body irradiation-based CD34-selected HSCT or monthly cyclophosphamide pulses. In this study, there was remarkably low treatment-related mortality in the HSCT group, which might be explained by more stringent inclusion criteria at patient’s selection. Patients in the HSCT arm had a higher risk of severe infections in the early stages, but after a median of 54 months of follow up, the HSCT-group showed significant benefit in term of clinical outcomes compared to the control group [14]. HSCT effects in SSc patients were also evaluated at a molecular level by Assassi and colleagues comparing whole blood transcript and serum protein levels between patients in the HSCT arm and in the cyclophosphamide arm of the SCOT trial. The authors focused on a specific molecular signature of SSc, including high interferon level, high neutrophil gene expression profile and low cytotoxic/NK profile, and reported a significant amelioration of all parameters in the HSCT group, suggesting that HSCT may “correct” SSc-related dysfunction at a deeper level as compared to cyclophosphamide group. Interestingly, these molecular changes reflected improvement in pulmonary and skin involvement [27]. 

All these studies impressively demonstrated that sustained clinical improvements over years after HSCT are achievable, which led autologous HSCT to be acknowledged as standard of care for refractory SSc. However, careful patient selection is crucial for identifying subjects who might benefit from HSCT without major complications. In this regard, severe cardiopulmonary dysfunction at the time of transplant was identified as a major risk for treatment-related mortality, suggesting that candidate screening through right heart catheterization, cardiac imaging, and pulmonary function testing is recommended, and that a patient-tailored treatment should be adopted in high-risk categories [28,29].

More recently, results from a non-interventional prospective study with 80 included SSc patients demonstrated a 2-year progression-free survival of 81% after autologous HSCT. By multivariate analysis, higher baseline skin-modified Rodnan skin score and older age at transplant have been identified as predictors for lower progression-free survival, whereas graft manipulation CD34-selection was associated with superior responses [30]. The necessity of CD34 graft selection is controversial. Some studies reported improved clinical outcomes and better progression-free survival of CD34-selected transplants compared to unmanipulated transplants, with comparable toxicity and infection rate [30,31], while other failed to show any significant difference [32].

By May 2022, 776 patients with SSc have been treated with HSCT within the EBMT registry (Figure 1). While autologous HSCT is considered to be a standard treatment option [33], use of allogeneic HSCT for SSc remains anecdotal due to higher risk of transplant-related complications [34,35,36,37]. Reduced-toxicity HSCT platforms are needed to further investigate and expand allogeneic procedure in AD patients [38].

## 3. Adoptive Cellular Therapies 

Alongside HSCT, other innovative therapeutic strategies with immunomodulatory properties have been investigated in the context of autoimmune disease, including SSc [12] (Figure 2).

## 4. Mesenchymal Stem Cells (MSC)

Mesenchymal stem cells, a heterogeneous population of stromal cells with high regenerative capacity that can be isolated, cultured, and expanded ex vivo, represent a promising source for cellular therapy approaches targeting SSc due to their immunosuppressive, angiogenic and anti-fibrotic properties [39]. Bone marrow (BM) stroma has been the main source for MSCs for decades; however, it became evident that MSCs with similar biological properties can also be isolated from placenta, Wharton’s jelly, blood vessels, dental pulp, derma and adipose tissue. The clinical interest in MSCs arises from their ability to differentiate toward mesodermal cell lineage, including chondroblasts, adipocytes and osteoblasts, and from their paracrine effects exerted through the secretion of trophic factors, cytokines, and other bioactive molecules within extracellular vesicles. MSCs plasticity and their regenerative and immunomodulatory properties are known to be driven by micro-environmental factors. This cellular plasticity reflects also on surfaces’ markers expression, which changes overtime due to senescence, inflammation, micro-environment changes and other pathological conditions. The International Society of Cellular Therapy (ISCT) has released guidelines regarding the minimum criteria that need to be met to define MSCs, which include multi-lineage differentiation potency, capacity to adhere to plastic surfaces, positive expression of CD105, CD73 and CD90 surface markers and negative expression of CD45, CD34, CD14, CD19 and HLA-DR surface markers [40].

MSCs of different tissue origin have been investigated for treating several indications, including tissue injuries, autoimmune diseases, and metastatic cancer [41,42,43,44,45,46]. Use of both autologous and allogeneic MSCs has been investigated. Given the low HLA expression on MSCs, there is not HLA-matching restriction when searching from MSC donors, minimizing the risk for immunological reactions [47]. However, recent studies reported some innate and adaptive immune responses triggered by allogeneic MSCs administration [48,49] despite being less potent compared to responses triggered by other cell lines. 

Indeed, the destiny and long-term persistence of infused MSCs are a matter of debate. In preclinical models, syngeneic MSCs can persist for a long time, whereas allogeneic MSCs tend to be eliminated more rapidly [50]. Eggenhofer et al. observed viable MSCs only within the lungs after intravenous infusion, but not in other tissues [51]. Finally, monitoring of infused unmanipulated MSCs is challenging, due to a lack of cell-specific markers; a combination of cytokines’ hyper expression and inhibition above certain limits has been proposed as an indirect way to monitor MSCs potency and activity. In allogeneic setting, quantitative PCR chimerism can be potentially adopted for MSCs tracking. 

Interestingly, the clinical effects of MSC-based therapy can often be observed for longer periods compared to their persistence in vivo. In a study on six patients with osteogenesis imperfecta treated with gene-marked MSCs, Horwitz et al. documented increased bone mineral density and reduced bone fractures despite only less than 2% of MSCs actually engrafted [52]. In an experimental model of SSc, Maria et al. showed how anti-fibrotic effect documented after MSCs infusion lasted up to 21 days, despite MSCs being undetectable after 7 days [53]. Similar results were shown in other settings, including myocardial infarction [54] and cerebral ischemia [55]; in the latter, MSCs injection determined prolonged benefits despite the majority of MSCs failing to engraft and only a minority differentiating into astrocytes. 

This discrepancy has led to the hypothesis that MSCs’ biological effects do not rely only on cell engraftment and differentiation, but also on paracrine effects exerted through the secretion of trophic factors, cytokines and other bioactive molecules. For example, autologous MSCs collected from patients with SSc appear to increase levels of pro-angiogenic factors in vitro [56] and to promote revascularization in vivo [57] despite studies documenting their defective differentiation into endothelial cells [58]. In recent years, major interest has grown around MSCs-derived exosome vesicles, a relevant part of MSCs’ secretome that contain proteins, DNA, mRNA, and miRNA, which can be released in the extracellular environment and horizontally transferred to target cells [59]. Through these and other mechanisms, MSCs promote chemoattraction, cellular survival and growth, tissue repair. Furthermore, Bartholomew and colleagues initially reported that MSCs can influence immune system function by suppressing lymphocytes response and preventing in vivo rejection of skin graft. Subsequent experiments showed that MSCs shape the immune system by favoring the shift towards a regulatory T cell phenotype, inhibiting dendritic cell maturation and natural killer (NK) cells’ cytotoxicity, and downregulating B-cell proliferation. Notably, MSCs’ immunomodulatory properties depend on the inflammatory status, again confirming the susceptibility to the surrounding micro-environment. Indeed, some [60,61], but not all authors [62,63], reported that MSCs collected from patients with autoimmune diseases, despite showing similar differentiating potential, phenotype and surface markers expression compared to MSCs from healthy donors, display reduced clonogenic, proliferating and migrating capacity. A pro-fibrotic phenotype has also been reported in MSCs from these patients [64,65].

All these complex and fascinating characteristics have induced investigations towards MSCs as treatment for immune system dysregulation. A paradigmatic example is graft-versus-host disease [GvHD]. Since the first report of the successful use of third-party MSCs in a case of refractory acute GvHD [66], several phase I–II studies have shown the safety and applicability of this approach [67,68]. Despite some authors have raised concerns about higher infection rate [69] and relapse rate [70], subsequent studies did not confirm these observations. Remestemcel-L (Prochymal), an off-the-shelf BM-derived MSC product, has shown to be effective in combination with steroids as a first-line treatment for acute GvHD [71]; in the steroid-refractory GvHD setting, Remestemcel-L as a single agent led to 61% of overall response and significant improvement of survival outcomes [72]. MSCs role in GVHD prophylaxis is anecdotal, with controversial results; whereas some authors reported faster engraftment [73], decrease in acute GvHD rate [73,74,75] and chronic GvHD rate [76] in patients receiving prophylactic peri-transplant MSC infusion, others failed to observe any significant improvements [77,78]. Non-homogeneity of MSCs subtypes, dose and timing might partially explain these observations.

### 4.1. Intravenous MSCs Use

In animal models of SSc, several in vivo studies showed that the infusion of MSCs could limit the cellular damage and collagen deposition. For example, bone marrow-derived MSCs infused intravenously after exposure to bleomycin in a rat model of SSc resulted in decreased levels of tissue injury markers within the bronchoalveolar lavage fluid and decreased levels of proinflammatory cytokines [79]. Similarly, Moodley et al. reported improvement of pulmonary fibrosis after the infusion of umbilical cord-derived MSCs in a bleomycin-induced SSc model, with reduced levels of TGF-beta and IFN-gamma, as well as decreased collagen deposition within the lungs [80]. Subsequently, Yang et al. documented improvements in cutaneous fibrosis after the infusion of umbilical cord-derived MSCs in the a bleomycin-induced SSc model, with decreased collagen synthesis and inhibition of Th-17 cell function [81]. Analogous results have been obtained in the HOCI-induced model, in which the elevated levels of plasmatic nitric oxide and of cutaneous/lung tissue α-SMA and TGF-β1 normalized after umbilical cord- or bone marrow-derived MSC infusions, reaching nearly a normal histopathology of lung and skin [82,83,84] Interestingly, also the administration of extracellular vesicles derived from MSCs primed with IFN-gamma has been demonstrated to improve lung-fibrosis in preclinical studies [85,86]. 

Given these experiences in preclinical models, MSCs have been investigated in patients with SSc in recent decades. Published data are restricted to small case series or retrospective studies, but all reported high safety of the treatment with a very low rate of therapy-related mortality (Table 1) [57,87,88]. 

In a retrospective study, Liang et al. investigated the safety of a single dose of intravenous allogeneic MSCs from either a bone marrow or cord blood source; the study focused on 404 patients with autoimmune diseases treated from 2007 and 2016, including 39 cases with SSc. The primary endpoint was safety and tolerability of the treatment. The five-year overall survival was about 90%, with disease-progression or disease-related complication being the most frequent cause of death, and only one patient experiencing therapy-related death. The infection rate was up to 30%, with severe infection being around 13%, which, according to the authors, could not only be explained as a treatment complication, but might have been partially explained by disease-related immune dysfunction [90].

Zhang et al. investigated the combination of plasmapheresis and allogeneic MSC transplant in 14 patients with SSc. As per study design, patients received repeated plasmapheresis, with subsequent three cyclophosphamide pulse and a single dose of 1 × 10^6^ cells/kg of body weight of MSC. Efficacy was measured through serologic testing and organ function analysis. The authors observed significant improvements in the mean modified Rodnan skin score after one year of follow up, as well as in pulmonary function tests in a subset of patients with lung fibrosis. Levels of anti-Scl70 autoantibodies, VEGF and TGF-beta decreased overtime [91].

More recently, Farge et al. reported the use of bone marrow-derived MSCs in 20 patients with insufficient response or contraindications to immunosuppressive therapy or HSCT. MSCs-infused doses were either 1 × 10^6^ or 3 × 10^6^ per body weight kilogram. The infusion was overall well tolerated, without treatment-related severe adverse events. Fifteen patients responded, with clinical benefits in terms of skin thickness [92]. 

Small case series have been reported as well, all showing at least transitory clinical improvements [57,88]. Optimal cell doses, administration timing, as well as source (autologous versus allogeneic) are still a matter of debate, and large, prospective clinical studies are warranted. Van Rhijn-Brouwer [93] reported the opening of a randomized, placebo-controlled study proposal (MANUS trial, NCT03211793) using intramuscular MSC injection in 20 patients suffering from SSc with DU refractory to standard treatments. Other studies have been posted on clinicaltrials.gov, aiming to investigate allogeneic MSC infusion in this setting (NCT05016804, NCT04432545, NCT04356287, NCT02213705).

### 4.2. Loco-Regional MSCs Use

Loco-regional MSCs use may have the advantage of treating specific SSc manifestations that severely impact quality of life, like ischaemic ulcers and skin fibrosis, in a targeted way. A promising MSCs source for local treatments appears to be the adipose tissue (AT), a tissue enriched in MSCs 500-fold greater than BM, AT-MSCs that are more easily available with painfulness collection procedures and minimal ethical considerations. AT-MSCs can be collected through liposuction from subcutaneous AT of abdominal wall or AT biopsy and expanded in vitro [83]. The harvested AT is composed of mature adipocytes (90%), extracellular matrix, and a stromal vascular fraction (SVF) which consists of AT-MSCs, along with endothelial progenitor cells, immune cells, fibroblasts, smooth muscle cells, mature endothelial cells, pericytes and cells not characterized yet [94]. SVF contains a percentage of MSCs estimated at 2–10% [95].

The regenerative potential of AT-MSCs is dependent on the paracrine effects of their secretome, constituted by a wide range of chemokines, cytokines, and protein growth factors: prostaglandin 2, vascular endothelial growth factor (VEGF) and interleukin (IL)-4, IL-6, IL-10 and IL-1 receptor antagonist [96,97]. All together, these soluble mediators sustain angiogenesis and tissue remodeling and suppress local inflammatory responses based on the local environment. Indeed, hypoxic conditions enhance a pro-angiogenic profile of AT-MSCs [98,99]. To obtain SVF, some authors used the commercially available Cytori therapeutics Celution800/CRS device (Cytori Therapeutics, San Diego, CA, USA) that processes a fat aspirate to remove fat and lipids and extract SVF or, as referred to by the company, “adipose derived regenerative cells (ADRC)”. The advantage of this technique is that removal of fat cells allows for the injection of the processed SVF or ADRC directly into arteries without risk of fat embolism. However, in order to reduce the time and costs of carrying out fat grafting without a cell lab, other authors centrifuged the harvested fat tissue and eliminated the upper oily supernatant as well as blood and debris at the bottom of the centrifuge. Only the middle layer containing adipose-derived cell fractions, such as AT-MSCs and endothelial progenitor cells, is then used for loco-regional use. 

AT-MSCs have already been applied to regenerate tissues in many diseases like cardiovascular diseases, breast reconstruction after radiation and burn injuries [100,101,102]. In SSc, AT-MSCs have been used locally to treat skin fibrosis and DU, both responsible for important disability and morbidity. First studies have been conducted in localized scleroderma lesions, i.e., linear scleroderma and morphea, and showed an improvement in skin elasticity and appearance [103]. Granel et al. described a significant improvement in hand skin elasticity and function [104], later confirmed by other authors [105,106,107]. In addition, several authors reported improved labial rhyme opening through reduced perioral fibrosis and neoproliferation of dermal capillaries [108,109,110,111,112]. More recently, two controlled trials failed to reach the primary endpoint of significant improvement of hand function in SSc patients treated with SVF [113,114]. Larger trials are needed to better understand the therapeutic efficacy of AT-MSCs injection on skin fibrosis of SSc patients. Really, the micro-environment of the site of injection may have compromised the regenerative capacity of AT-MSCs or induced a myo-fibroblast-like differentiation; in addition, the use of SVF instead of whole fat may have contributed. Indeed, it is thought that SVF may be superior, since adipocytes and other cells are eliminated leaving higher concentrations of MSCs, but clear demonstration is lacking [115].

The pro-angiogenic properties of AT-MSCs have been exploited through subcutaneous injection in patients affected by DU resistant to standard treatment, with different protocols and wide variations in the harvesting, processing, and injection techniques. The use of AT-MSCs as DU therapy has been encouraged by clinical data obtained from intramuscular injection of BM-MSCs in 49 patients with DU enrolled in five different studies [116,117,118,119,120].

Bank et al. [121] were the first authors to treat with autologous fat grafting primary and secondary Raynaud’s phenomenon. In this way, the authors have observed a significant decrease in DU with minor side effects (transient numbness, cellulitis responsive at antibiotics in 1 case). Subsequently, two pilot studies demonstrated the efficacy of autologous fat grafting in inducing ulcer healing when injected at the border of the larger ulcers or at the base of the corresponding finger. Del Bene et al. treated 9 SSc patients for a total of 15 ulcers, achieving the healing of 10 DU and the size reduction above 50% of 2 DU in 8–12 weeks, with pain improvement in almost all patients. Only DUs that were located in the lower extremity or associated with atherosclerosis did not heal [122]. Del Papa et al. treated 15 SSc patients with long-lasting DU, achieving healing in all the patients in 2–7 weeks and with a significant reduction of pain in a few weeks. Interestingly, they also observed an increase in nail-fold capillaries at capillaroscopy at 3 and 6 months, and a significant after-treatment reduction of digit artery resistivity measured by high-resolution echo-color-Doppler, strongly supporting the pro-angiogenic efficacy of regional autologous fat grafting [123]. 

Encouraged by these data, in order to overcome pilot study limits, a monocentric randomized controlled study has been conducted. SSc patients were randomly assigned to receive either autologous fat grafting (*n* = 25) or a sham procedure with saline solution injection (*n* = 13). DU healing occurred in 23/25 and 1/13 patients treated with fat grafting and sham procedure, respectively. The 12 patients who received the unsuccessful sham procedure subsequently underwent rescue autologous fat grafting with DU healing in all of the patients within 8 weeks. The authors also confirmed significant pain improvement and an increase in nailfold capillaries already observed in the pilot study [124]. 

These studies have demonstrated that AT-MSCs therapies based on subcutaneous injection are safe and efficacious in treating microvascular complications of SSc such as long-lasting DU and could be the milestone for future clinical trials and applications.

## 5. Regulatory T Cells and Chimeric Antigen Receptor T Cells

Regulatory T cells (Tregs), a subtype of CD4 T helper cells with immune suppressive properties, are dysfunctional in several autoimmune syndromes [125]. The frequency of circulating Tregs was found to be inversely correlated with the disease activity and severity of SSc patients [126]. Theoretically, administering Tregs in this population can potentially lead to more tolerogenic micro-environment. Kamio and colleagues reported fibrosis regression after the infusion of splenic-derived Tregs in an animal model of bleomycin-induced pulmonary fibrosis [127]. Initial studies in humans using T regs (or IL-2, with subsequent boost of T regs) have focused on GvHD prophylaxis and treatment in allogeneic transplant recipients [128,129,130,131]. Despite being safe overall, polyclonal Tregs mediated sub-optimal responses in initial clinical trials, mainly due to the low amount of disease-relevant antigen-specific T cells [132,133]. A phase I/II clinical trial is currently being conducted to evaluate the tolerability and efficacy of autologous Tregs in patients with SSc (NCT05214014).

Chimeric antigen receptor (CAR) T cells are an engineered cellular product that combine B-cell antibody-based antigen recognition with T-cell cytotoxicity. This technology is redefining therapeutic strategies in the onco-hematologic field; however, the CAR ability of conferring new antigen-specificity while boosting cellular activation can be potentially employed in other setting. Indeed, in autoimmune diseases, the abrogation of autoreactive cells obtained through CAR T cells may be more profound and prolonged and might overcome the inaccessibility of inflamed tissues compared to standard drug-based approach.

In preclinical studies on models of autoimmunity, Ellebrecht et al. reported chimeric autoantibody receptors (CAARs), in which the engineered chimeric receptor displays the target protein of autoantibodies in its extracellular domain; through this mechanism, CAAR T cell would specifically bind the B-cell receptors of an autoreactive B cell, triggering its apoptosis and reducing its autoantibody production [134]. Different autoantibodies described in SSc appear to be associated with different clinical presentation and might be investigated for developing SSc-specific CAAR T cells. Whereas in some autoimmune disease specific antigen targets can be identified (e.g., insulin in type 1 diabetes), in other subtypes, including systemic sclerosis, the lack of one single autoantigen represents an obstacle for CAR-T cells’ efficacy. 

CAR-T cells are also being studied as an anti-fibrotic approach. An interesting preclinical analysis published by Aghajanian and colleagues reported the use of CAR-T cells directed against fibroblast activation protein in a mouse model of cardiac fibrosis. In this model, CAR-T CD8+ cells successfully ablated cardiac fibroblast expressing xenogeneic antigen [135]. This approach has been further explored by the same group through the adoption of transient anti-fibrotic CAR-T cells in vivo by delivering modified messenger RNA in T cell–targeted lipid nanoparticles in order to improve its safety [136].

Despite these promising results in pre-clinical models, clinical trials in patients with connective tissue diseases are still preliminary. Following the positive results of the first CAR-T-cell therapy in systemic lupus erythematosus [125], the authors recently presented an update of five patients with multi-organ involvement and insufficient response to multiple previous therapies [137]. All patients received autologous CD19-directed CAR-T cells, without developing relevant toxicities, and only three patients experienced fever (cytokine-release syndrome grade 1), one received a single infusion of tocilizumab with symptoms relief; no infection occurred. After a follow-up of three months, circulating CAR-T cells were still detectable, along with a re-emergence of B cells after a median 110 of days post-infusion. All patients had a normalization of serum double-stranded DNA antibodies and complement levels and, most importantly, the resolution of nephritis and other disease-related symptoms. These data suggest a rapid response of autoimmune disease to CAR-T-cell therapy, although follow up is needed to determine long-term efficacy. 

Major concerns regarding CD19-directed CAR-T cells is prolonged B-cell depletion and the subsequent increase of infection rate, which is already consistently high in population autoimmune diseases. In this regard, alternative approaches include T regs, a potential platform for CAR construct engineering with less B-cell-depleting capacity and more immune-modulating properties. CAR-T regs might potentially target either autoreactive B-cell markers (e.g., CD19), or antigens expressed on the tissue under autoimmune attack, or pathologic immune-complexes [138]. Elinav and colleagues firstly reported the use of CAR-T regs in a murine model of autoimmune colitis, showing their ability to specifically modulate pathologic effector cells, improving colitis-associated signs [139]. Other authors have investigated CAR-T regs in autoimmune models: Blat et al. reported decreased severity of ulcerative colitis after infusion of carcinoembryonic antigen-specific CAR-T regs [140], whereas Tenspolde et al. reported safety and long-term in vivo persistence of insulin-specific CAR-T regs in type 1 diabetes mellitus [141]. Compared to unmanipulated T regs, potential advantages of CAR-T reg technology include the ability to redirect T regs’ suppressive capacities while increasing the number of antigen-specific cells to be transferred to the patient. Compared to other CAR-T products, CAR T regs might have a higher safety profile in terms of off-target toxicity and effect on immune competence [138].

Ongoing studies using CAR-T cells directed against CD19 and B-cell maturation antigen (BCMA) in patients with SSc are available on clinicaltrials.gov (NCT05085444). 

## 6. Tolerogenic Dendritic Cells 

The surrounding micro-environment can skew the maturation path of immature dendritic cells toward either an immunogenic or a tolerogenic, semi-mature phenotype. Tolerogenic dendritic cells (tolDCs) play central roles in maintaining peripheral tolerance homeostasis, through secretion of anti-inflammatory cytokines (e.g., IL-10) and the expression of inhibitory molecules during antigen presentation. Indeed, after cell-to-cell interaction between T cells and tolDC, the absence of co-stimulatory molecules on tolDC prevents T-cell activation, whereas the presence of inhibitory molecules induces T-cell anergy or differentiation toward T regs. 

Many protocols for in vitro and in vivo generation of tolDCs from human monocytes have been investigated, mainly in animal models. The characterization of tolDC phenotype has been reported by Comi and colleagues [142]. In this context, tolDCs loaded with disease-specific target antigens have the potential to reset and re-educate the dysfunctional immune system by abrogating pathological autoreactive T cells and promoting their anergy or differentiation toward regulatory phenotype. In a murine model of multiple sclerosis, Mansilla and colleagues reported that infusion of tolDCs loaded with myelin oligodendrocyte glycoprotein increased regulatory T cells population and decreased autoreactive T cells’ clones, which translated into neurological improvements [143]. After infusion, tolDCs have been traced to several organs, including lungs, kidney, liver, spleen, lymph nodes, thymus, bone marrow and central nervous system. However, different generation platforms as well as route of administration appear to influence tolDCs anti-inflammatory properties, their expression of trafficking chemokine receptors and their migration capacity into target organs [144,145].

TolDCs have been investigated in phase I studies in patients with multiple sclerosis, autoimmune arthritis, Crohn’s disease, and type 1 diabetes [146]; these preliminary analyses proved that tolDC administration was overall safe, determined a trend of decrease of pro-inflammatory cytokines and T cells and increase of T regs, as observed in murine models, and improved clinical symptoms. However, standardization of protocols for tolDC generation and administration is needed before their routine clinical use is possible, as well as better knowledge of immune monitoring to measure in vivo biological effect. No studies on tolDCs-based therapy for SSc have been published yet. 

## 7. Open Questions and Future Perspectives

The biological complexity and the clinical heterogeneity of SSc raise several challenges to finding the optimal treatment (Figure 3). Autoreactive B- and T-cell clones and the dysregulation of adaptive and innate immune systems play central roles in the disease pathogenesis. Several therapeutic approaches to target them have been investigated; however, as of now, there are no biologic or immunosuppressive drugs able to effectively improve long-term drug-free survival [147]. 

The lack of effective therapies has guided several approaches toward the development of cell-based therapies able to restore self-tolerance [10]. Autologous hematopoietic stem cell transplant has been investigated in SSc patients in order to obtain profound immune ablation of self-reactive cells. Several phase III trials proved its benefit over standard immunosuppressive drugs in terms of long-term disease control. However, patients with severe cardiopulmonary dysfunction who undergo HSCT are at higher risk of transplant-related morbidity and mortality, thus, we could speculate that HSCT should be offered at earlier stages of their disease before the development of cardiopulmonary impairment. The appropriate patient selection and the investigation of patient-tailored, low intensity conditioning regimen represent important steps in the transplant algorithm for SSc.

Efforts have been made to investigate other non-transplant, cell-based therapies. Despite the increasing knowledge of SSc physiology, it is still unclear which patients would not have sustained response from immunosuppressants and thus could benefit from early use of cellular-based treatments. Among all, mesenchymal stem cells have raised significant interest: along with their tolerogenic properties, MSC-based treatments showed an interesting potential as regenerative therapies, able to repair tissues already damaged, without conferring the potential toxicities secondary to conditioning regimen and aplasia of HSCT. Both local and systemic use of MSCs have been studied in phase I and II studies, documenting their overall safety and potential efficacy in stabilizing, or improving, the disease. Systemic therapies may have application in diffuse cutaneous thickening and internal organ fibrosis such as interstitial lung disease, one of the most frequent causes of mortality in SSc patients and an exclusion criterion for HSCT, when extended. Local MSCs therapies have been applied in digital ulcers and skin fibrosis (mainly perioral) with improvement of both vascularization and elasticity. These techniques could be an add-on therapy to standard drug treatments in refractory disease. MSCs exert their regenerative properties also trough their secretome, that prolong their biological effect in situ. More studies are needed in order to evaluate the persistence of MSCs engrafted or infused and the long-term effects of these cells in SSc patients. 

Another emerging potential approach could be the use of T-cell therapies. The use of polyclonal T regulatory cells is intriguing as, theoretically, restoration of this cellular subset which is dysfunctional in patients with SSc might provide immune modulation and disease control. Unfortunately, challenges in in vitro expansion and long-term persistence, as well as difficulties in the identification of specific antigens, have significantly slowed T-regs introduction in clinical fields. Results from ongoing studies are awaited. 

Engineering of T cells through the creation of chimeric antigen receptors has led to substantial changes in the treatment of onco-hematologic diseases. CAR-T cells have the advantage of recognizing antigen in an HLA-independent fashion. Targeting B-cell-specific antigens like CD19 and B-cell maturation antigen might determine a broad depletion of autoreactive B cells, also within affected tissues, thus resetting the autoimmune process. However, these benefits might be counterbalanced by profound normal B-cell depletion, as seen in hematologic patients receiving CAR-T cells, leading to prolonged infectious risk. Recent reports of patients with systemic lupus erythematosus receiving CD19-CAR-T-cell therapy are promising, showing sustained disease remission and low toxicity [137,148]. However, more clinical studies are warranted to determine the long-term safety in patients with autoimmune diseases. Using T regs as substrate for CAR-T cells might be safer in this context, as they do not have a cytotoxic profile and could potentially restore immunologic tolerance. Choice of alternative CAR-T-cell target is also a matter of debate: fibroblasts are a potential targetable pathway, as shown by Aghajanian et al. [135] in murine models of cardiac fibrosis, although it is unclear, given the complex pathophysiology of SSc, if targeting only the final steps of the biological process could translate into sustained clinical resolution. Nevertheless, the CAR-T-cell strategy could potentially be similarly effective as autologous HSCT, yet with less toxicity, and might be a valid alternative in frail patients who cannot undergo HSCT. Also, modulation of chemotherapy dosages in the lymphodepletion regimen is being evaluated in onco-hematologic patients receiving CAR-T cells and could translate into a less toxic preparative regimen.

In conclusion, at this stage, more data are needed to draw conclusions regarding which patients might benefit from which approach. Treatment for SSc is an evolving field, and new approaches are currently under evaluation. Better understanding of the pathologic mechanisms will lead to the development of new, specific therapies to target cellular interactions and to impact clinical outcomes.

## Figures and Tables

**Figure 1 cells-11-03346-f001:**
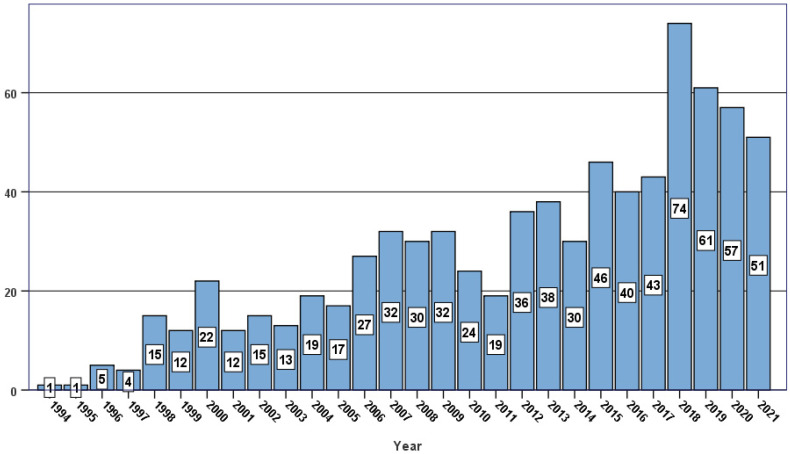
Number of autologous transplant procedures for the treatment of systemic sclerosis. Data as reported to the European Blood and Marrow Transplantation Society (EBMT).

**Figure 2 cells-11-03346-f002:**
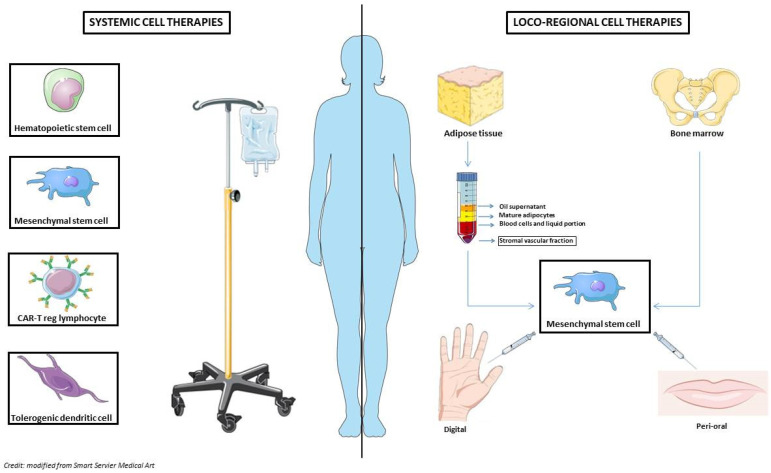
Cellular therapies might be adopted for both systemic and loco-regional use in patients with SSc. On the left side: systemic treatments including hematopoietic stem cell transplantation, mesenchymal stem cells, CAR-T cells and tolerogenic dendritic cells. On the right side: local treatments rely on mesenchymal stem cells obtained from adipose tissue or bone marrow successively engrafted to digits or perioral area.

**Figure 3 cells-11-03346-f003:**
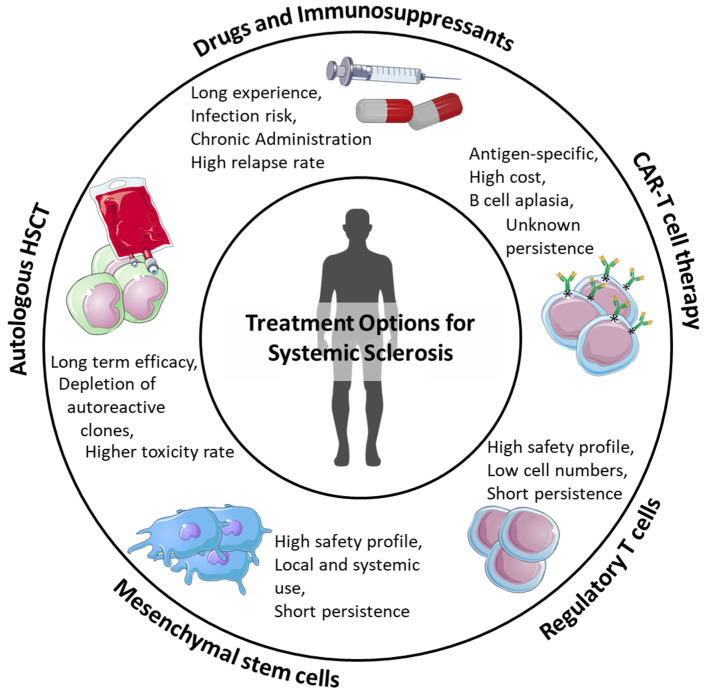
Advantages and limitations of different types of available or potential treatment options for SSc.

**Table 1 cells-11-03346-t001:** Clinical trials investigating mesenchymal cells for the treatment of systemic sclerosis.

MSC Isolation	Administration Route	Source	Treated Patients	Clinical Target	Outcome	Major Adverse Events	References
BM-derived MSCs	Intravenous	Allo	1	Severe diffuse SSc	Ulcer healing and improved skin elasticity/vascularization	None	Christopeit et al., 2008
BM-derived MSCs	Intravenous	Allo	5	Severe diffuse SSc	All patients had temporary clinical improvement	Minor respiratory infection	Keyzer et al., 2011
BM-derived MSCs	Intravenous	Allo	20	Severe diffuse SSc	15 patients showed sustained improvement in skin thickness	None	Farge et al., 2022
BM-derived MSCs and peripheral blood MSCs	Intramuscular injection	Auto	6 (4 SSc, 2 mixed connective tissue disease)	Severe ischaemia and necrosis in fingers and/or toes	Pain relief in 5/6 patients	None	Kamata et al., 2007
BM-derived MSCs and peripheral blood MSCs	Intramuscular injection	Auto	2	Severe digital and malleolar ulcers	All had improvements with ulcers healing, pain relief, reduction of RP	None	Nevskaya et al., 2009
BM-derived MSCs and peripheral blood MSCs	Intramuscular injection	Auto	46 (24 SSc, 22 other connective tissue diseases)	Severe digital ulcers	20/23 SSc patients had improvement in pain and ulcers	None	Takahashi et al., 2009
BM-derived MSCs	Intramuscular injection	Auto	8	Severe digital ulcers	All had ulcers size and pain improvement	None	Ishigatsubo et al., 2010
BM-derived MSCs	Intramuscular injection	Auto	40 (11 SSc, 29 with arteriosclerosis obliterans)	Severe digital ulcers	All had pain and trans-cutaneous oxygen tension improvement	Major limb amputation due to pre-existing osteomyelitis	Takagi et al., 2014
Adipose derived cell fractions	Subcutaneous injection	Auto	13	Raynaud’s phenomenon	10 patients had clinical benefit, 3 reported no changes.	None	Bank et al., 2014
Adipose derived cell fractions	Subcutaneous injection	Auto	20	Peri-oral fibrosis	All patients had improved skin elasticity and vascularization	Small areas of ecchymosis	Del Papa et al., 2015
Adipose derived cell fractions	Subcutaneous injection	Auto	15	Severe digital ulcers	All patients displayed clinical benefit with fast healing of digital ulcers	None	Del Papa et al., 2015
Adipose derived cell fractions plus platelet-rich plasma	Subcutaneous injection	Auto	6	Peri-oral fibrosis	All patients had improved skin elasticity and vascularization	None	Virzì et al., 2017
Adipose derived cell fractions	Subcutaneaous/perioral injection	Auto	6	Skin scleroderma	Improvement in 4 patients, stabilization in all	None	Scuderi et al., 2013
Adipose derived cell fractions	Subcutaneous injection	Auto	5	Peri-oral fibrosis	All patients had improvement to perioral fibrosis	None	Onesti et al., 2016
Adipose derived cell fractions plus platelet-rich plasma	Subcutaneous injection	Auto	7	Peri-oral fibrosis	All patients had improvement to perioral fibrosis	None	Blezien et al., 2017
Adipose derived cell fractions	Subcutaneous injection	Auto	62	Peri-oral fibrosis	Improvement in mouth opening	Superficial wound infection	Almadori et al., 2019
SVF	Subcutaneous injection	Auto	12	Severe hand dysfunction	Improvement of pain, grasping capacity, finger oedema, Raynaud’s phenomenom, quality of life	None	Guillaume-Jugnot et al., 2016, Daumas et al., 2017, Granel et al., 2015
SVF	Subcutaneous injection	Auto	18	Severe hand dysfunction	Improvement of skin fibrosis, hand oedema, and quality of life	None	Park et al., 2020
Adipose derived cell fractions	Subcutaneous injection	Auto	9	Severe digital ulcers	All patients had pain improvement, digital ulcers improvement or healing	None	Del Bene et al., 2014
Adipose derived cell fractions vs placebo	Subcutaneous injection	Auto	25 vs. 13	Severe digital ulcers	23/25 and 1/13 patients had digital ulcers improvement and healing, pain reduction and improvement on nail fold capillaroscopy	None	Del Papa et al., 2019
SVF vs placebo	Subcutaneous injection	Auto	20 vs. 20	Severe hand dysfunction	Improvement of hand function in both groups, with no superiority of the SVF	Hypoxaemia during the surgical process	Daumas et al., 2022
Adipose derived cell fractions vs placebo	Subcutaneous injection	Auto	48 vs. 40	Severe hand dysfunction	No improvement of hand function	Aspiration pneumonia, hypotension, angina	Khanna et al., 2022

*BM, bone marrow; MSCs, mesenchymal stem cells; SVF, stromal vascular fraction; S*
*Sc, systemic sclerosis. Adapted from Rozier et al. [89].*

## Data Availability

Not applicable.

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
