# Peer review of "Cellular-Based Therapies in Systemic Sclerosis: From Hematopoietic Stem Cell Transplant to Innovative Approaches"

_cells, 2022, doi:10.3390/cells11213346_

Round 1

Reviewer 1 Report

The review is timely, comprehensive and well written other than minor grammatical and spelling errors which should be corrected. 

Two major concerns:

1/ What is the justification for considering fat grafting as a form of "mesenchymal stem cell therapy"  other than adipose tissue is relatively enriched in MSC / there is no data presented showing the content of MSC in the adipose tissue preparations used in the studies indicated . Pursuant to this data which should be made clear, the Table describing forms of "MSC" should be divided into studies using bone fide MSC and those using cellular preparations containing other cell types.

2/In figure 3 CAR T cells are described as non HLA dependent. This is only partly correct as the CAR T cells still retain native TCR which are HLA restricted and indeed are thus required to be formed from autologous peripheral blood/

Author Response

Two major concerns:

1/ What is the justification for considering fat grafting as a form of "mesenchymal stem cell therapy"  other than adipose tissue is relatively enriched in MSC / there is no data presented showing the content of MSC in the adipose tissue preparations used in the studies indicated . Pursuant to this data which should be made clear, the Table describing forms of "MSC" should be divided into studies using bone fide MSC and those using cellular preparations containing other cell types.

Answer: Thank to the reviewer for the interesting comments and suggestions. Fat grafting is considered a form of loco-regional mesenchymal stem cells (MSC) therapy since harvested adipose tissue is composed of mature adipocytes (90%), extracellular matrix, and a stromal vascular fraction (SVF) that is composed of MSCs, along with endothelial progenitor cells, immune cells, fibroblasts, smooth muscle cells, mature endothelial cells, pericytes and cells not characterized yet (Gimble J.M., et al. Circ Res. 2007). SVF contains a percentage of MSCs estimated at 2-10% (Bourin P. et al. Cytotherapy 2013). Currently, the most important rationale in clinical use of fat grafting lies in the presence of MSCs in the SVF, cells that are characterized by self-renewal properties, multi-lineage differentiation potential, strong proliferation, and migration abilities in vitro. All these characteristics make adipose tissue-derived MSCs desirable candidates for regenerative therapeutic protocols.

In the studies cited in this review, the authors obtained MSCs from abdominal wall liposuction according to Coleman’s procedure (Coleman S.R. Clin Plast Surg 1997). Other authors used the commercially available Cytori therapeutics Celution800/CRS device (Granel B. et al. Ann Rheum Dis. 2015). The device processes the fat aspirate to remove fat and lipids and a mixture of cells termed SVF or as referred to by the company adipose derived regenerative cells (ADRC) are available for reinjection. The advantage of this technique is that the removal of fat cells allows injection of the processed SVF or ADRC directly into arteries such as the coronary artery without risk of fat embolism. However, in order to reduce time and costs carrying out fat grafting without a cell lab, other authors centrifuged the harvested fat tissue and eliminated the upper oily supernatant as well as blood and debris at the bottom of the centrifuged. Only the middle layer containing adipose derived cell fractions such as stromal MSCs and endothelial progenitor cells, was locally injected (Del Papa N. et al. Arthritis Res Ther. 2019). We defined in the present review the intermediate layer as “adipose derived cell fractions” instead of simple “adipose tissue” just to underline a better differentiation from SVF. Few sentences have been added from line 296 to 315 to clarify the content of MSCs in adipose tissue. Table 1 and references have been updated.

2/In figure 3 CAR T cells are described as non HLA dependent. This is only partly correct as the CAR T cells still retain native TCR which are HLA restricted and indeed are thus required to be formed from autologous peripheral blood/

Answer: We thank the reviewer for the comment; we removed the statement “HLA-independent” in the figure to avoid confusion and improve the readability.

Reviewer 2 Report

-This is a well written, sound manuscript that describes in detail all pertinent literature regarding cellular-based therapy in SSc. The authors also discuss experimental treatments that are being used for other diseases but that would have potential in SSc. I consider this is pertinent.

-The discussion about HSCT is pertinent, but I would encourage the authors to include the interesting information regarding the molecular signatures that evolve differently in patients that underwent HSCT vs those who received CFM in the SCOT trial (Assasi S et al. Ann Rheum Dis 2019; 78:1371-1378).

-The manuscript is very informative and the figures are adequate. I think that the manuscript should be accepted after minor review.

-I have no further comments

Author Response

-The discussion about HSCT is pertinent, but I would encourage the authors to include the interesting information regarding the molecular signatures that evolve differently in patients that underwent HSCT vs those who received CFM in the SCOT trial (Assasi S et al. Ann Rheum Dis 2019; 78:1371-1378).

Answer: We thank the reviewer for the suggestion, we added the following paragraph in the HSCT chapter.

“HSCT effects in SSc patients were also evaluated at a molecular level by Assassi and colleagues comparing whole blood transcript and serum protein levels between patients in the HSCT arm and in the cyclophosphamide arm of the SCOT trial. The authors focused on specific molecular signature of SSc, including high interferon level, high neutrophil gene expression profile and low cytotoxic/NK profile, and reported a significant amelioration of all parameters in the HSCT group, suggesting that HSCT may “correct” SSc-related dysfunction at a deeper level as compared to cyclophosphamide group. Interestingly, these molecular changes reflected improvement in pulmonary and skin involvement (27). “

 -The manuscript is very informative and the figures are adequate. I think that the manuscript should be accepted after minor review.

-I have no further comments

Answer: We thank the reviewer for the comments.

Reviewer 3 Report

The review paper summarizes cellular therapies in SSc, from pre-clinical models to clinical applications, paving the way for more sophisticated cellular therapies such as mesenchymal stem cells, regulatory T cells, and even CAR-T cell therapies. The paper is well organized and written, however, there are several concerns and suggestions:

Minor concerns:

1.     Line 175 “MSCs of different tissue origin have been investigated for treating several indications……. can author include a few studies here

2.     Line 190 “Interestingly, the clinical effects of MSC-based therapy can be often observed for longer 191 periods compared to their persistence in vivo” can author few examples in the form of studies

3.     In figure 3 authors should increase the resolution of the picture and font size of the inside text for better clarity

Author Response

  1. Line 175 “MSCs of different tissue origin have been investigated for treating several indications……. can author include a few studies here

Answer: We thank the reviewer for this comment, we added some relevant studies both in clinical and preclinical setting, and some reviews in the field.

  1. Line 190 “Interestingly, the clinical effects of MSC-based therapy can be often observed for longer 191 periods compared to their persistence in vivo” can author few examples in the form of studies

Answer: We added some relevant studies in the field.

“In a study on six patients with osteogenesis imperfecta treated with gene marked MSCs, Horwitz et al. documented increased bone mineral density and reduced bone fractures despite only less than 2% of MSCs actually engrafted (52). In an experimental model of SSc, Maria et al. showed how anti-fibrotic effect documented after MSCs infusion lasted up to 21 days, despite MSCs were undetectable after 7 days (53). Similar results were shown in other settings, including myocardial infarction (54) and cerebral ischemia (55); in the latter, MSCs injection determined prolonged benefits despite majority of MSCs failed to engraft and only a minority differentiate into astrocytes.”

  1. In figure 3 authors should increase the resolution of the picture and font size of the inside text for better clarity

Answer: We modified the figure accordingly.